# Recent Advances in Liver Tissue Engineering as an Alternative and Complementary Approach for Liver Transplantation

**Dileep G. Nair * and Ralf Weiskirchen ***

Institute of Molecular Pathobiochemistry, Experimental Gene Therapy and Clinical Chemistry (IFMPEGKC), Rheinisch-Westfälische Technische Hochschule (RWTH) University Hospital Aachen, D-52074 Aachen, Germany
* Correspondence: grdileep@gmail.com (D.G.N.); rweiskirchen@ukaachen.de (R.W.)

**Abstract:** Acute and chronic liver diseases cause significant morbidity and mortality worldwide, affecting millions of people. Liver transplantation is the primary intervention method, replacing a non-functional liver with a functional one. However, the field of liver transplantation faces challenges such as donor shortage, postoperative complications, immune rejection, and ethical problems. Consequently, there is an urgent need for alternative therapies that can complement traditional transplantation or serve as an alternative method. In this review, we explore the potential of liver tissue engineering as a supplementary approach to liver transplantation, offering benefits to patients with severe liver dysfunctions.

**Keywords:** liver tissue engineering; tumor necrosis factor-α; interleukin-6; hepatocyte growth factor; insulin-like growth factor; platelet-derived growth factor; transforming growth factor-β

## 1. Introduction

The liver is the largest gland in the human body, mediating essential functions in homeostasis, metabolism, serum protein production, storage of glycogen, drug detoxification, immune system, and production and secretion of bile acids [1]. The liver has a lobular structure and is divided into three major zones: zone I supplies oxygen and nutrients to the hepatocytes, zone II connects the zones, and zone III performs glycogenesis, detoxification, and lipogenesis functions [2]. These diverse functions are made possible by the synchronized functioning of different cell types. The functional unit of the liver is a lobule, hexagonal in shape (Figure 1) [2].

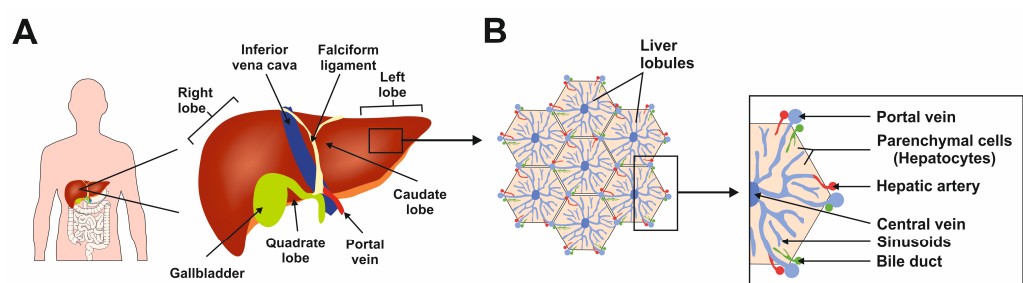

**Figure 1.** The complexity of human liver architecture. (**A**) The liver is located in the upper right-hand portion of the abdominal cavity on the top of the stomach and intestines. (**B**) The liver is composed of microscopic units called lobules, which are roughly hexagonal in shape and comprise rows of hepatocytes radiating out from a central point. This figure was redrawn in a modified form from [3].

Hepatocytes are parenchymal cells and constitute in majority to the lobule. The non-parenchymal liver cells are hepatic stellate cells (HSCs), Kupffer cells (KCs), and liver sinusoidal endothelial cells (LSECs) [2]. Most of the homeostasis functions mentioned above are supported mainly by the hepatocytes. The functions of KCs, HSCs, and LSECs

are in hepatic immunity, storing vitamin A and lipids, and control of barrier function with immune homeostasis, respectively [2]. The presence of HSCs present in the liver supports the organ's unique ability to regenerate under certain conditions [2].

Drugs, viruses, carcinomas, hemochromatosis, and Wilson disease affect the normal function of the liver [2]. Some of the diseases are life-threatening diseases, for which a liver transplant is the only resolution (Figure 2).

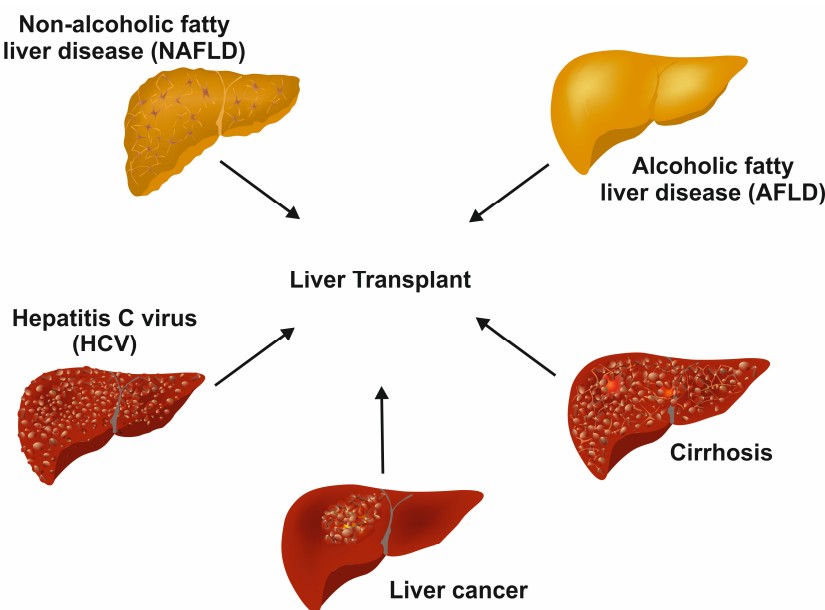

**Figure 2.** Terminal diseases requiring liver transplants as the last option. In the long term, different untreated liver diseases, such as chronic hepatitis C infection, non-alcoholic fatty liver disease, alcoholic fatty liver disease, decompensated cirrhosis, and liver cancer, result in end-stage liver disease requiring liver transplant.

In developed countries, non-alcoholic fatty liver disease (NAFLD) is the most common chronic liver disease [4]. An excess accumulation of fat in the hepatocytes promotes steatosis and ultimately leads to hepatocyte apoptosis [5]. Portal and lobular inflammation and fibrosis may follow with the development of cirrhosis with accumulated collagen and altered live architecture [5]. NAFLD is mostly tied to insulin resistance, obesity, and metabolic syndrome [5].

Hepatitis C virus (HCV) is one of the major forms of liver hepatitis. A recent review by Taha and colleagues mentioned that there are approximately 1.5 million new infections per year, according to the World Health Organization (WHO) [6]. The virus is spread through exposure to blood; it multiplies once it enters the hepatocytes by receptor-mediated endocytosis. In the majority of infected persons, the condition becomes lethargic, resulting in progressing liver damage [4]. The virus can cause hepatitis in both chronic and acute forms. The resulting life-long cirrhosis or hepatocellular carcinoma (HCC) that develops at later disease stages is the leading cause of HCV-associated death. Unfortunately, HCV infections are not detectable without testing. In the early stages, the infection has no noticeable symptoms, which makes it complicated to interfere with disease progression. Also, there is no vaccine currently available against HCV.

HCC is the most common type of liver cancer and the fourth most common reason for cancer-related deaths worldwide [7]. Hepatitis B virus (HBV) or HCV infections contribute to most of the HCC conditions worldwide [7]. The HCC condition leads to cirrhosis for most of the patients with a reduction in hepatocyte proliferation, fibrosis, and detrimental effects on other liver cells, ultimately leading to cancerous nodule formation [8]. Metabolic disease conditions, including NAFLD, obesity, and overweight, are shown to be associated

with an increase in HCC [7]. The common therapies for early stages of the HCC are resection, ablation, and liver transplantation [7].

Biliary atresia is a major disorder affecting infants and one of the diseases benefiting from liver transplantation [9]. Its etiology is due to the obstruction of bile flow leading to cholangiopathy during fetal and prenatal stages [9]. Recent studies have identified the role of genetic contributions along with the previously known causes, including viral infections and toxins [9].

Wilson's disease is an inherited disorder affecting copper metabolism, leading to liver damage and neurological disturbance [10]. It is an autosomal recessive disorder affecting the gene *ATP7B*, encoding a hepatic plasma membrane copper-transporting protein [10]. The accumulation of copper in the liver results in mild hepatitis or even cirrhosis. The symptoms are visible, gradually following the accumulation of copper in organs like the liver and brain [4].

All liver diseases and conditions discussed above lead to cirrhosis, which is characterized by the presence of fibrotic septa, which can lead to portal hypertension and terminal liver disease (Figure 3) [11].

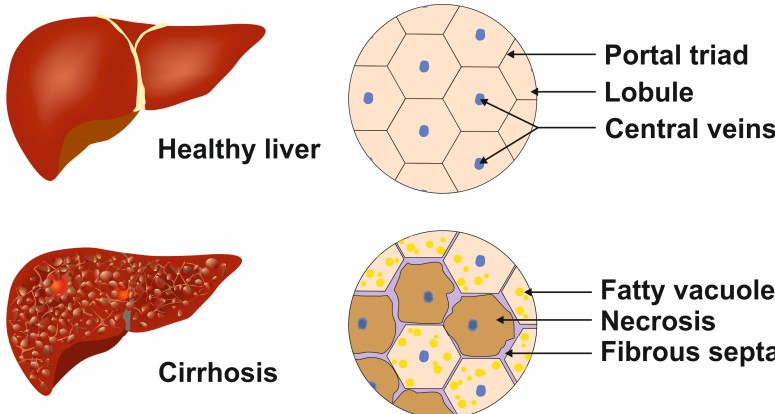

**Figure 3.** Cellular phenotypic changes during terminal liver disease. In the healthy liver, the lobules contain regular central veins and peripheral portal triads, while in the cirrhotic liver, the uniform structure is destroyed by the incorporation of fatty vacuoles, fibrous septa, and necrotic areas. These alterations lead to portal hypertension.

Several cell types contribute to the development and progression of cirrhosis [12]. During inflammation or injuries and in the presence of cytokines and growth factors like platelet-derived growth factor (PDGF) and transforming growth factor-β (TGF-β), HSCs proliferate and migrate, similar to intestinal smooth muscle cells [13]. Further, they transform into a myofibroblast phenotype generating collagen and extracellular matrix (ECM), which is the hallmark of fibrosis [12]. The changes in LSECs include defenestration and capillarization along with interleukin 33 (IL-33) secretion, which further activates HSCs, ultimately leading to the progression of fibrosis [12]. Once activated by injurious factors, KCs damage hepatocytes by secreting soluble mediators [12]. Liver fibrosis is reported to be enhanced by KC-mediated liver inflammation [12]. Hepatocytes, consisting of ~70% of the liver cell mass, play a major role in fibrosis and cirrhosis. During inflammation or phases of liver damage, hepatocytes are prone to apoptosis, thereby releasing additonal factors promoting fibrogenesis and cirrhosis. Even with the unique regenerative potential of the liver, several diseases or damages, as explained above, push to stages beyond the point at which repair of affected liver tissue is possible.

## 2. Liver Transplantation: Significance and Current Status

For patients with acute liver failure or last-stage liver diseases, liver transplantation (LT) is the final treatment option. It involves the removal of a damaged liver (nonfunctional), which is replaced by either a part of a healthy liver from a living donor or a

complete liver from a deceased donor [2]. Recently, there has been good progress in the field of LT. However, the shortage of donors is the major limitation [2]. The waiting list of LT candidates has been consistently high (~10,000 persons) in the US [14]. Moreover, the LT procedure may be associated with complications and extensive anti-rejection medication requirements. The second major issue in LT is the unsatisfying long-term outcomes of the LT recipients. The 20-year survival rate is only approximately 50%, potentially due to the complications of renal failure and adverse side effects induced by the long-term immuno-suppression that liver transplant recipients receive [15,16]. Also, there are disparities still prevalent in LT based on socioeconomic conditions, race/ethnicity, geographic area, and age of transplanted patients [14].

Alternative therapies are beneficial to address the scarcity of donors, surplus demand for LTs, and high risks or after-effects involved. Even though there are recent advances in identifying biomarkers or molecular diagnostic tools for predicting organ rejection after LT, further studies are needed to obtain a conclusive picture [17]. Recent advances in the field of regenerative medicine, stem cells, and tissue engineering can serve as a potential alternative or supplement to the LT process.

Here in this review, we discuss the progress in the liver tissue engineering (LTE) field and highlight the potential of customized LTE to support patients.

## 3. Liver Tissue Engineering: Principles and Progress

The ultimate goal of LTE is to restore partial or total function of the liver during liver failure. A fully functional liver is the ultimate aim of LTE, and functional liver tissue can be used for drug testing [2,18]. Moreover, LTE has the potential to develop an extracorporeal liver support (ECLS) system performing the essential functions of the liver to reduce mortality or to bridge a patient to a liver transplant [19]. With recent advances in the field of regenerative medicine, tissue engineering holds high potential to progress as an alternative or supplement to LT. A major advance in the LTE area was published by Chhabra and colleagues, reporting the development of a vascularized liver model to understand liver regeneration [20]. In their study, they showed that the inclusion of endothelium-lined channels to fluid flow in the 3D liver platform improved liver regeneration [20]. Song et al. reported successful transplantation of engineered 3D co-aggregates of human-induced pluripotent stem cell (iPSC)-derived hepatocyte-like cells (iPS-H) encapsulated in biocompatible hydrogel capsules [21]. Moreover, Bhandari et al. demonstrated that co-cultures with 3T3 fibroblasts enhance survival and reduce rapid de-differentiation of rat hepatocytes [22]. The incorporation of engineering technologies like dielectrophoretic (DEP) force improved the speed, handling and label-free precise cell patterning for mimicking the lobular architecture of the liver [23]. Selden et al. demonstrated that encapsulation of liver cell spheroids in alginate beads proliferated in a bioreactor when introduced to pigs with irreversible ischemic liver failure retained liver functions [24]. This study further demonstrates that scaling up spheroid generation is possible with transport and retention of in vivo function [24]. Robert et al. reported bridging liver failure with donor hepatocytes in a patient who fully recovered thereafter [25]. Nevertheless, further studies on the performance and adaptability of tissue-engineered liver after transplantation are imperative.

Identification of ideal cell source, hepatocyte phenotype maintenance and vascularization are the major challenges faced by the LTE field. As discussed above, hepatocytes constitute the majority of the cell population in the liver. Stem cell technology has been explored as a source for diverse hepatic cells. However, low differentiation efficiencies, complicated procedures, and underlying carcinogenic risks are presently the technology's hurdles to overcome [18]. One of the major factors critical for LTE is the creation of a proper hepatic ECM. ECM has a major role in maintaining the structure and shape of the organ along with regulating cellular functions, while serving as a natural microenvironment. ECM constitutes ~3% of the relative area, with collagens I, III, IV, and V as the most abundant matrix proteins [26]. The difficulty in maintaining the phenotypes of the hepatocytes and other primary cells in compatible ECM is another major issue faced

by LTE [18]. Hepatocytes show a flattened-extended morphology with high cytoskeletal proteins and a high proliferation rate when cultured in a two-dimensional (2D) format. An ideal ECM facilitates cell-to-cell and cell-to-matrix interactions along with maintaining the cellular phenotype [18]. Apart from the need for an ideal ECM, direct contact between different hepatic cell populations is required for proper functioning [18].

### 3.1. Focus on Extracellular Matrix: Use of Biodegradable Biomaterials

Tissue engineering using scaffolds with biodegradable properties and structures is the most popular method and is capable of maintaining three-dimensional (3D) cell growth, tissue regeneration and overall function of the engineered liver tissue [26]. Since the liver is one of the organs with the most vascularization, it is ideal to have a biomaterial compatible with supporting vascularization. The scaffold can be designed to gradually degrade, leaving behind only the regenerated liver tissue once the engineered liver tissue matures and becomes functional. The major biophysical factors critical for ideal cell transplantation during tissue regeneration are (i) porosity, which generally affects cell adhesion, proliferation, migration, and differentiation; (ii) stiffness, also affecting the hepatocellular phenotypic properties; and (iii) geometry, directly influencing the seeded cells [2].

Commonly used biomaterials addressing the requirements for the hepatic cell types and ECM are mainly natural hydrogels and synthetic materials [2]. Hydrogels like alginate, chitosan, and gelatin have advantageous effects for promoting and restoring cell growth and function [8]. Marine polysaccharides and marine-derived chitosan show protective effects against liver damage while serving as scaffolds for tissue engineering [27]. The ECM of the liver contains collagens, which are the most abundant component. Several studies carried out using modified collagen for liver tissue engineering showed positive outcomes with the expression of hepatic markers of cell spheroids [26]. Hyaluronic acid, a commonly used matrix component that binds to CD44, is expressed by immature and mature hepatocytes. Hepatocytes that are formed as cell aggregates in hyaluronan scaffolds remained viable and proliferative active for more than 4 weeks [28]. Matrigel contains a mixture of proteins from murine chondrosarcoma composed of laminin, collagen type IV, proteoglycan, and heparan sulfate [29]. Several LTE model studies have used Matrigel as a scaffold for culturing hepatocytes and fostering stem cell differentiation. However, inconsistent mechanical properties, degradability, and lack of regenerative ability, along with potential xenogeneic and tumorigenic origin, are some of the shortcomings in the usage of these natural biomaterials, especially for supporting LTE for clinical applications [1]. Synthetic materials containing biodegradable polymers like polylactic acid, polyanhydrides, poly-L-lactic acid, and polycarbonates have more superior properties and support regeneration, transplantation, and biodegradation [30]. By modifying these synthetic biomaterials by incorporating proteins or bioactive domains, biocompatibility can be improved in these synthetic biomaterials [1]. Decellularized ECM has the advantages of compatibility and degradability compared to other natural or synthetic media, which will be discussed in more detail later in this review.

### 3.2. Knowledge from Studies on Post-Hepatectomy Liver Failures

Understanding the impact of the cytokine and growth factor signaling pathways involved in post-hepatectomy can be beneficial for optimizing LTE. Along with optimizing biomaterial scaffold, a proper balance of these factors can be recreated in in vitro conditions to improve the regenerative potential of the liver. Hoffmann et al., reported an elaborative study where modulation of several potential cytokines and growth factors were analyzed from 3353 articles, including around 1000 animal studies and a double number of human studies [22,31]. In this article, a number of potential predictive biomarkers and their implications for liver diseases were identified and discussed [31]. A list of cytokines/growth factors relevant to LTE are listed (Table 1) [32]. Incorporation of high-throughput screening

studies using ex vivo regenerative liver models can be useful to further validate the biomarkers identified.

**Table 1.** Potential biomarkers in liver diseases relevant to liver tissue engineering.

| Regeneration Stage | Growth Factors/ Cytokines | Major Roles | Enhances Regeneration |
| --- | --- | --- | --- |
| Priming stage | TNF-α, IL-6 | Essential for liver regeneration | Yes |
| Proliferation stage | PDGF-Rα<br>HGF<br>IGF | Replaceable with EGFR<br>Essential for liver regeneration<br>Liver growth, development, and regeneration | Yes<br>Yes<br>Yes |
| Termination stage | TGF-β | DNA synthesis inhibition in hepatocytes, ECM remodeling | No |

This table was adapted from [32]. Abbreviations used: ECM, extracellular matrix; HGF, hepatocyte growth factor; IGF, insulin-like growth factor; IL-6, interleukin 6; PDGF, platelet-derived growth factor; TGF-β, transforming growth factor β; TNF-α, tumor necrosis factor-α.

### 3.3. Whole-Organ Bioengineering Approach for Liver Tissue Engineering

Whole-organ bioengineering involves gentle decellularization using detergent solutions to retain the dimensional ECM with preserved architecture, including vasculature [33]. The native framework after decellularization can ultimately be used for successful organ transplants. This strategy is highly beneficial to utilize organs destined for human transplantation but discarded because of diverse reasons [34].

Whole-liver bioengineering has been in focus among LTE approaches [35]. Diffusion and perfusion techniques have been tested for liver acellularization [36]. Baptista and colleagues reported that the decellularization of the liver matrix showed favorable conditions for the survival of human fetal liver cells and human umbilical vein endothelial cells [37]. However, one of the major challenges in whole-liver bioengineering is repopulating the scaffold using appropriate cell type and source. Induced pluripotent stem cells were tested to produce functional hepatocytes in mice [38]. Further research on growing diverse primary liver cells to repopulate liver scaffolds, including hepatocytes, HSCs, and LSECs, is urgently needed. Once optimized, this powerful technique can preserve liver architecture, liver-specific ECM, and vasculature for cell survival and function with a multitude of applications, as shown in Figure 4.

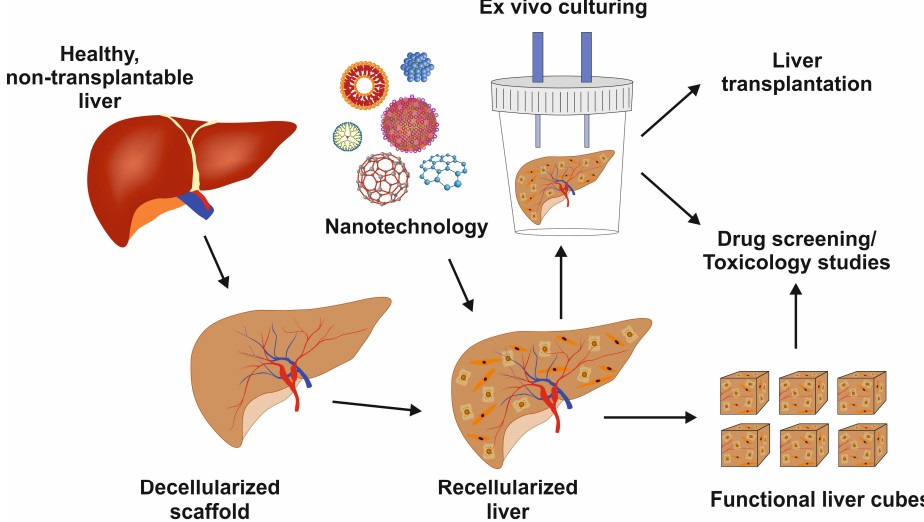

**Figure 4.** Image showing the decellularized liver scaffold, recellularization with nanotechnology support, and its potential uses for liver transplantation, drug development, or toxicology studies.

### 3.4. Recellularization in Liver Tissue Engineering

Repopulating the scaffold, either biomaterial-derived or decellularized, is complex as it involves parenchymal spaces, macrovascular lining, and biliary tree [39]. Parenchymal repopulation can be carried out using variable cell sources cultured in adequate quantities in an ideal environment, including hepatoblastoma cells like HepG2 cells. Primary hepatocytes were also tested in animal models, showing high functionality [40]. They have the advantage that they have, in clinical application, a much lower tumorigenic potential than the immortalized cell line HepG2 [41]. It has been further reported that rat primary hepatocytes stay viable for 14 days in decellularized Wistar rat livers with the capacity to express typical lineage markers [39,41].

Primary hepatocytes serve as a potential alternative to mesenchymal, embryonic, or induced pluripotent stem cell-derived hepatic cells [42]. The potential advantages of these cells include availability, a more mature phenotype, and a consistent cell source. Under physiological conditions, hepatocytes remain quiescent and proceed to the $G_1$ phase upon liver damage, which leads to their proliferation. Understanding hepatocyte proliferation is of utmost importance as it constitutes the majority of the functional liver. Elchaninov et al. demonstrated in rat models that there is delayed hepatocyte proliferation following subtotal hepatectomy [43]. Additionally, there is a delicate balance between pro- and anti-mitotic paracrine factors that determine their proliferation status [43]. The induction of Sox9 transcription factor, a major indicator for hepatocyte proliferation, and two genes encoding tumor necrosis factor-like cytokine TWEAK (TNFSF12) and its receptor Fn14 (TNFRSF12A) in the management of hepatocyte and cholangiocyte proliferation were reported [44]. However, with optimized protocols, stem cell-derived hepatocytes can be mass-produced with customized functionalities. Human mesenchymal stem cells (hMSCs) can be modified to support liver functions. During their culturing in in vivo conditions, they retained the expression of hepatic markers compared to their 2D cultures [34]. Human-induced pluripotent stem cells (hiPSCs) can be reprogrammed into several cell types and differentiate effectively into hepatocyte-like cells [45]. One major bottleneck in using iPSC-derived hepatocytes is the time taken for the entire culturing and differentiation to ultimately establish functional hepatocytes [45]. Recently, the group of Kamishibahara and colleagues reported optimized conditions using a Lamininin-511 (LN) direct coating and specific factors to differentiate hiPSCs into hepatocytes, mesodermal cells, endothelial cells (EC), and mesenchymal cells [46]. Liver organoids derived from these differentiated cells showed consistent hepatic functions [46]. Kajiwara and colleagues demonstrated that the variations in hepatic differentiation were mainly determined by the donor differences rather than the derivation methods [47]. Hence, autologous sourcing of hiPSC-derived liver cells can have an advantage. There are successful studies using differentiated hepatocytes from iPSC expressing hepatocyte markers and functioning in decellularized rat liver scaffolds [48]. Cell sourcing and culturing strategies for vascular tracts and bile ducts is another major area of LTE for which advanced studies to develop efficient strategies are ongoing [45]. Repopulating the decellularized liver may be a better approach with scaffolds in place after decellularization.

The long-term efficiency of using these cell sources or co-cultures for repopulating biomaterial/decellularized scaffolds is yet to be determined. In particular, the ideal culture conditions are shown to be more in a complex or 3D format, mimicking the in vivo environment. Some of the studies where functional cells were successfully cultured will be discussed in the next session.

### 3.5. Liver Cell Culture Methods

There are several ways reported to grow functionally differentiated liver cells. Yang et al. recently extensively reviewed various methods for biomedical applications. Some of the reported bottom-up tissue culture methods include micromodel co-culture, spherical aggregate culture, and cell sheet culture methods [45]. Nanofiber scaffold,3D bioprinting,

and decellularization-cellularization methods are additional top-down tissue engineering methods [45].

However, with reduced complications in requirements for the vascular cells or other ECM requirements, decellularized scaffolds can be advantageous. Park et al. used iPSC-derived porcine hepatocytes to populate decellularized porcine liver scaffolds [49]. The recellularized liver cells showed hepatic markers and functionality after culturing using a continuous perfusion system. Further, these recellularized scaffolds were successfully transplanted into rats, highlighting the potential of this approach [49]. A high-shear stress oscillation-decellularization method was employed by Mazza et al. to generate human acellular liver tissue cubes (ALTCs) [50]. These ALTCs were later seeded with functional liver cells derived from human parenchymal and non-parenchymal liver cell lines and human umbilical vein endothelial cells (HUVECs) [50].

## 4. Major Challenges in the Field of Liver Tissue Engineering

One of the major challenges in LTE is the availability of cell types and maintaining the functional engineered cells by recreating the in vivo conditions. Commonly explored cell sources include reversibly immortalized human hepatocytes and the usage of stem cell-derived hepatocytes [51]. Cell-cell interactions and cell-matrix interactions are yet to be characterized thoroughly to develop an efficient and sustainable microenvironment [51]. Several approaches, like microprinting of liver micro-organs, have been initiated to understand the inter-cellular interactions, ideal microenvironment, and other in vivo biological mechanisms [52]. LTE, being in the research stage of development, the scalability and cost are other major bottlenecks requiring future consideration. The requirement for an efficient bile transport system is another important and complicated puzzle that needs further research and development. Given the high significance of ECM, further characterization or modification to suit the enhanced and suitable functioning of the matrix is beneficial. Some of the potentially useful ECM variants or approaches are discussed in the following section.

The potential clinical complications that arise after the transplantation and long-term metabolic function requirements of the tissue-engineered liver are challenges requiring focus. The immunogenicity of the decellularized scaffolds and biomaterials has been reported for tissue-engineered organs [53]. Damage-associated molecular patterns (DAMPs) like cell/ECM damages, reactive oxygen species (ROS), and ECM components generally initiate the inflammation process [53,54]. In the case of decellularization, the usage of different reagents like ionic detergents and enzymes is shown to have DAMP release and ECM alteration effects and may result in complications after transplantation [55]. It has been reported that low molecular weight HAs released from ECM damage contributed to inflammation and graft rejection in lung transplantation [56]. In cardiovascular clinical space, incomplete decellularization of xenogeneic porcine heart valve implants resulted in severe adverse effects and, ultimately, the death of three out of four transplanted patients [57]. Therefore, further focus on the clinical side of tissue engineering is required to reduce immunogenicity and, hence, improve the adaptability and functionality of tissue-engineered organs.

## 5. Future Directions of Liver Tissue Engineering

With the advancements in sourcing in vivo aligned cells, scaffolds to sustain their proper functioning and maintaining proper ECM, LTE holds an important role. However, there are several potential strategies that can further improve LTE efficiency. Some of the potential approaches are discussed in the following.

### 5.1. Usage of Anti-Inflammatory Biomaterials

The cell-biomaterial interface plays a crucial role in determining the tissue-regeneration success rate [58]. An inflammatory response during the initial stages of biomaterial implantation helps in tissue repair and regeneration, whereas it provokes a detrimental impact at later phases [58]. It has been demonstrated that the use of immune-modulatory biomaterials can be highly useful in modulating the inflammatory response and, hence, improving

oral wound healing and tissue regeneration [58]. Similar approaches can be practical using immune-interactive biomaterials for LTE. The physiochemical and biological properties of the biomaterials can be adjusted to lean toward anti-inflammatory pathways.

### 5.2. Personalized Approaches for Liver Tissue Engineering

Patient-specific approaches may involve using patient-derived cells combined with biomaterials to create engineered liver tissue. This approach mitigates the possibility of donor rejection and helps to recreate conditions that match the patient's genetic makeup and medical history [59]. If necessary, the use of advanced genome editing technologies like zinc-finger nuclease (ZFN) genome editing, transcription activator-like effector nuclease (TALEN) driven genome editing, and CRISPR-Cas9 can be employed to correct or modify hepatic cells before integrated into the LTE platform for future transplantation [60,61].

### 5.3. Progress in the Liver-on-a-Chip Approach

There is enormous progress in the field of organ-on-a-chip technology, which combines cell culture models and microfluidics. These advanced models possess the highest relevance to the physiological conditions including cell-to-cell, cell-to-matrix interactions, flow of oxygen and signaling molecules, low stiffness environment, and multicellular structures [62].

With more advances in the field of ideal cell culture sources, compatible readout technologies for efficient real-time detection of liver or drug metabolites may ultimately yield high-throughput screening platforms at low cost for studying hepatology and drug development. Liver-on-a-chip with biosensors facilitating automation and live cell imaging can support their generalized usage in the future [63]. Further, progress toward introducing multiple organs working in conjunction with the liver in an integrated manner and the introduction of immune system components may lead to better-equipped models for alignment with human biology.

### 5.4. Role of Artificial Intelligence

There has been incredible progress during the last years in the field of implementing computational models for analyzing images of tissue sections for disease progress predictions. Lu and coworkers reported the usage of a histomorphometric-based image classifier of nuclear morphology to risk stratify squamous cell carcinoma patients [64]. Recently, Candita et al. reviewed the potential advantage of artificial intelligence in supporting radiologists to distinguish HCC (hepatocellular carcinoma) from other liver diseases [65].

Since LTE involves multiple cell types, specific requirements of variable growth factors at different growth stages, cell-cell interactions, and gradient of nutrients, oxygen, and ECM, artificial intelligence-enabled micro-robotic systems would be able to serve as a powerful tool to increase the overall security of LT and LTE.

### 5.5. Bioactive Molecules Delivery and Recapitulating In Vivo Settings

Along with serving as an ideal ECM, biomaterials can be modulated to provide bioactive molecules such as growth factors and nutrients in a controlled manner for cell growth and differentiation. They can also support vascularization using angiogenic molecules or with materials with texture-enhancing vascularization and for proper overall liver function [2]. However, adverse effects like leakage to healthy tissues through the bloodstream can occur if higher doses of growth factors are provided to the system [66]. An optimal biocompatible delivery system, either of natural or synthetic origin, that can precisely and safely release growth factors has high importance in LTE. A focus on biomaterials with similarity to ECM is useful to address several requirements. Chitosan and chitin derived from animals are well-characterized as substrates for tissue engineering. Chitosan nanoparticles are also explored as growth factor delivery systems and are reviewed in the section Advances in Nanotechnology Field in more detail. Hyaluronic acid is a major component of the extracellular matrix in vertebrates. Chemical modification of hyaluronic

acid can increase the affinity for growth factors [67]. Hyaluronic acid maintained the absorption and release of EGF from the scaffold targeting skin models. Further, animal models with skin wound models treated with hyaluronic acid and EGF healed well compared to EGF alone models [68]. Choi and colleagues demonstrated the usage of heparin and hyaluronic acid-derived scaffold to supply vascular endothelial growth factor (VEGF) with a heparin-binding domain and hepatocyte growth factor (HGF) resulting in a rapid and tight endothelium restoration [69]. Similarly, VEGF-dextran-poly(lactic-co-glycolic acid) (PGLA) microspheres were used by Zhang et al. to deliver VEGF for therapeutic neovascularization [70]. These vehicles promoted mature vessel formation in rat hind-limb ischemic models [66,70]. Biodegradable polymers can be tailored for mechanical properties and degradation kinetics [66]. Some of the common polymers include poly(glycolic acid) and poly(p-dioxanone). Hydrogels containing biodegradable polymers are well-explored in tissue engineering. Kobayashi et al. used hydrogel-based delivery of basic fibroblast growth factor (bFGF) to restore acute vocal fold scar in a canine model [71]. This method improved the availability and working time of bFGF, which is relatively short [71]. Kaminski used a layer-by-layer liposome to deliver EGF, which improved the sustained delivery of EGF by 5-fold, compared to one layered liposome [72]. Nonetheless, further studies using similar carrier technologies using biocompatible materials in hepatic systems are urgently needed to supplement LTE.

Maintaining hepatocyte polarity is a major challenge in LTE. Kim et al. demonstrated functional 3D hepatic tissue in vitro using a cell sheet stratification technology [73]. In their system, a sheet of hepatocytes was sandwiched between two endothelial sheets and transmission electron microscope (TEM) and immunocytochemistry techniques revealed the functionality of the artificial arrangement [73]. For the ideal functional phenotype of hepatocytes and hepatic sinusoidal cells, sheer stress induced by blood flow is essential. The inclusion of cell polarity and elasticity into biomaterials may help to establish transferrable in vitro models that are functional in in vivo settings as well [2].

### 5.6. Advances in Liver 3D Bioprinting

3D bioprinting technology has the potential to precisely blend biomaterials serving as the ECM and diverse cells to mimic liver physiology [74]. The major forms of 3D bioprinting are (i) extrusion-based, (ii) droplet-based, and (iii) photocuring-based bioprinting systems [74]. The HepaRG cell line containing hepatorganoids was successfully 3D bioprinted by Cuvellier and coworkers [75]. The hepatorganoids demonstrated various liver functions and, when transplanted to mice with liver damage, prolonged their survival time [75]. Further, a complex cell culture 3D bioprinted model using hepatocyte-like cells, HSC and HUVECs was established [76]. These landmark studies can help to understand liver functions and the development of pathological conditions like fibrosis. However, the current status of hepatic 3D bioprinting technology needs more refinement in areas like printing liver sinusoid and plate structures, minimizing the shear damage of cells in the extrusion-based methods, developing vascular system and biliary system in alignment with the liver system [76].

### 5.7. Advances in Imaging Techniques

Several technologies are being used to image liver regeneration after a transplant or surgical removal of a part of the liver in a noninvasive way [77]. Some of the common technologies are computed tomography (CT), which works on the differential absorbance of X-rays by the tissue; magnetic resonance imaging (MRI), where the time-variant magnetic spins of hydrogen atoms under a magnetic field are used; and ultrasound, which uses acoustic impedance [77]. Seyedpour et al. reported the biomechanic characteristics of liver regeneration using advanced MRI techniques where several fluidic properties, including blood volume, velocity, and sheer stress, were extracted [78]. However, all these technologies are directed toward clinical applications. Further development in advanced MRI or similar technologies that do not require room-shielding or other sophisticated

infrastructure could be highly beneficial to establish novel in vitro LTE models. These technologies, along with characterization at the lobule-level or cellular-level, supported by computational models, can help understand liver fibrosis and ways to prevent it. Ultimately, increased knowledge in these technologies will improve the field of LTE and the success rate of liver transplantation.

*5.8. Focus on the Clinical Aspects of LTE*

Along with efforts to improve the LTE, the focus needs to be given to identifying means to prevent the immunogenicity of the engineered liver for transplant. Further research on identifying immunogenic factors and biocompatible scaffolds can lead to successful transplantation [53]. As discussed above, the usage of bioactive materials as scaffolds with anti-inflammatory properties and the ability to supply ideal growth factors necessary for sustained organ performance will be useful. Elsewhere, several growth factors with carriers have been approved to improve tissue regeneration and proper functioning like Reranex® by Novartis uses PDGF-BB to heal diabetic foot ulcers, bone morphogenetic protein 2 (BMP-2) in a collagen carrier is provided as INFUSE® Bone Graft by Medtronic for fracture repair [79]. Future research on combining growth factors and biomaterials can benefit LTE and further trasplantation.

*5.9. Advances in Nanotechnology Field*

Recent advances in the field of nanotechnology hold the potential to LTE and transplant by supporting processes, including efficiently encapsulating and delivering growth factors. The flexibility with respect to the size, shape, characteristics, and specificity of nanoparticles makes them an efficient approach. Improved internalization, penetration, controlled drug release, and reduced adverse reactions make them more versatile [80]. In the case of liver diseases, several organic or inorganic nanoparticles (NPs) have been explored, including silicon-based nanoparticles, polymers, and metal nanoparticles (Figure 5) [81].

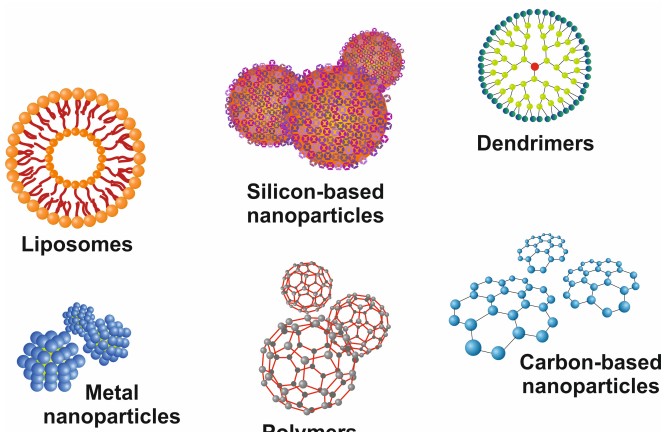

**Figure 5.** Representative images of commonly used nanoparticles. There are different classes of drug delivery devices available, including polymers, repetitively branched molecules (dendrimers), bilayered lipid vesicles (liposomes), paramagnetic iron-oxide nanoparticles, and nanoparticles composed of other metals, silicon, or carbon. All these nanocarriers have precise-branched structures and can be functionalized by covalently or non-covalently linked surface modifications.

The diversity in chemical composition and extensive physiochemical properties make them ideal candidates as therapeutic cargo agents for therapeutic nucleic acids, antibody fragments, aptamers, and small molecules (Figure 6) [81].

Elsewhere, the successful usage of chitosan nanoparticles as a dual growth factor delivery system was reported [82]. Similarly, the chitosan nanoparticles incorporated with epidermal growth factor (EGF) and fibroblast growth factor (FGF) showed increased fibroblast growth with poor inflammatory response [83]. Furthermore, chitosan nanoparticles

with incorporated HGF were used to differentiate murine bone marrow mesenchymal stem cells from hepatocytes [83]. These studies are highly promising and underpin the significance of nanoparticles as efficient cargoes in supporting LTE.

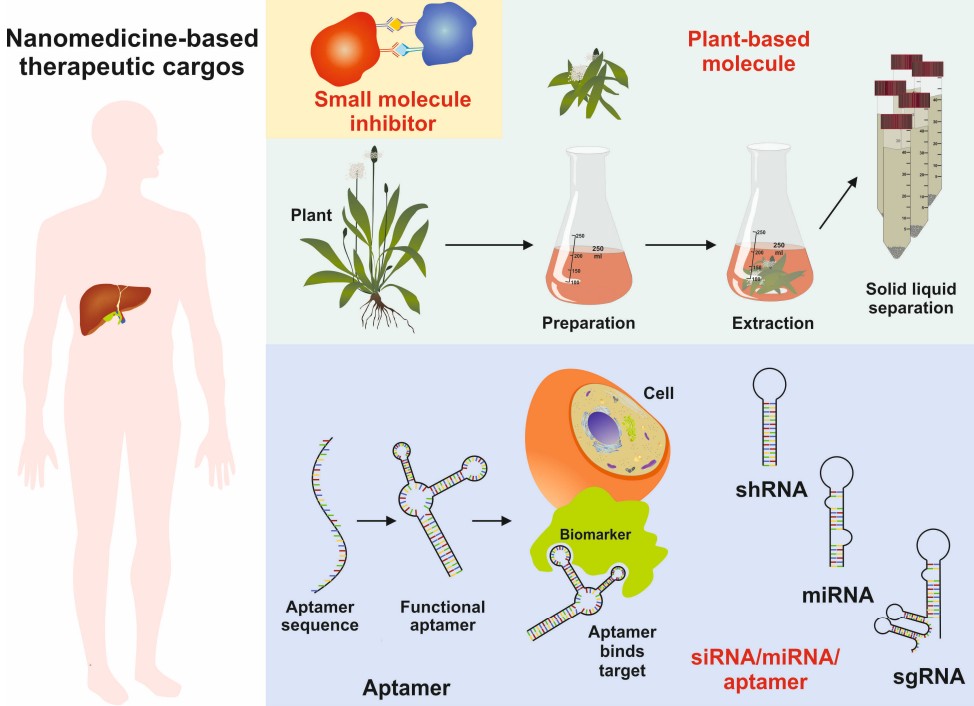

**Figure 6.** Image showing the usage of nanoparticles as therapeutic cargoes, including functional aptamers, miRNA, shRNA, sgRNA, and natural small molecules of plant origin.

## 5.10. Biosensors and Modeling Technologies

Monitoring the viability, function, and phenotype of embedded cells in a tissue-engineered organ is challenging. Current major monitoring techniques include imaging, fixing the samples, immunofluorescence, and other labor-intensive and invasive technologies. Noninvasive and label-free technologies like incorporating biosensors are advantageous to provide real-time and on-site monitoring of the system. Lee et al. demonstrated nanotechnology-based biosensors to monitor neural differentiation of stem cells, focusing on optical and electrochemical methods [84]. Wavelength-modulated differential laser photothermal radiometry (WM-DPTR) technology was used in the group Guo to detect serum glucose levels in human skin [85]. Similar technology can be used to monitor cell metabolism in an LTE model. Nanoprobes and microelectrodes were used to detect $H_2O_2$ and ATP levels, respectively in in vitro systems [86]. In LTE systems, the detection of albumin and urea, responses to P450 enzyme activity, and other readouts are advantageous to keep track of the hepatocyte performance.

Recently, mathematical and computational models have been developed to support bioreactors and microphysiological systems to optimize culture conditions and in vivo predictions [87]. The contributions include modeling of cell fate, cell interactions, nutrient supply and waste removal [87]. Kerkhofs et al. demonstrated a qualitative model of the differntiation network in chondrocyte maturation essential in endochondrial bone formation [88]. Emmert et al. used computational modeling to predict the long-term in vivo performance of tissue-engineered heart valves [89]. Khosravi and team reported a computational model of neovessel development that can define the contributions of immunobiological and mechanobiological processes in tissue-engineered vascular grafts [90]. A computational model to predict the drug release from electrospun nanofiber mats was reported by Milosevic et al. [91]. Adapting these models in LTE has the potential to contribute to the ongoing efforts to develop an efficient platform with long-term in vivo functioning.

## 6. Conclusions

In conclusion, LTE has great potential to circumvent the bottlenecks faced by the current treatment of acute or chronic liver diseases and liver transplantation. With the recent advances in the field of tissue engineering, sourcing of functional liver cells, and understanding of liver microenvironment, LTE can be progressed as a complementary/alternate strategy to liver transplant.

**Author Contributions:** Conceptualization, D.G.N.; resources, R.W.; data curation, D.G.N. and R.W.; writing—original draft preparation, D.G.N.; writing—review and editing, D.G.N. and R.W.; supervision, R.W.; funding acquisition, R.W. All authors have read and agreed to the published version of the manuscript.

**Funding:** R.W. is supported by the German Research Foundation (grants WE2554/13-1, WE2554/15-1, and WE2554/17-1), the Deutsche Krebshilfe (grant 70115581), and the Interdisciplinary Centre for Clinical Research within the Faculty of Medicine at the RWTH Aachen University (grant PTD 1-5). The funders had no role in the design of this article or in the decision to publish it.

**Institutional Review Board Statement:** Not applicable.

**Informed Consent Statement:** Not applicable.

**Data Availability Statement:** This review only presents data that were previously published. No new data were generated.

**Acknowledgments:** The authors are grateful to Sabine Weiskirchen (IFMPEGKC, RWTH University Hospital Aachen, Aachen, Germany) for preparing the Figures for this review.

**Conflicts of Interest:** The authors declare no conflicts of interest.

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
