# Peer review of "Recent Advances in Liver Tissue Engineering as an Alternative and Complementary Approach for Liver Transplantation"

_cimb, doi:10.3390/cimb46010018_

Round 1
Reviewer 1 Report
Comments and Suggestions for Authors
After a thorough review of your manuscript on Recent Advances in Liver Tissue Engineering as an Alternative and Complementary Approach for Liver Transplantation, I have the following observations to make:
-
Overgeneralizations and Lack of Precision: The manuscript sometimes makes broad statements without providing specific evidence or data to back up these claims. For instance, the paper broadly discusses the potential of various cell sources for liver tissue engineering, but fails to delve into the specific benefits and drawbacks of each cell source or comparative studies. The explanation and analysis need to be more precise and case-specific.
-
Oversights in Clinical Translation: The paper does a great job of reviewing the scientific aspects of liver tissue engineering, but it does not sufficiently address clinical implications. There needs to be some mention of how these tissue-engineered livers would function in vivo, the potential complications, and the current limitations in translating these technologies to the bedside. These considerations are key towards the clinical relevancy of the research.
-
-
Omission of Important Research: Some key research studies that have significantly contributed to the field of liver tissue engineering seem to be missed in this review. Including these could provide a more comprehensive overview.
-
-
Coherence: You maintained a good level of coherence throughout the paper, discussing the problems of liver diseases, potential solutions, and recent advances.
-
The review does not present any contrasting views or disagreements within the field. Including different perspectives or debates would give a more balanced view of the current state and future directions of liver tissue engineering.
-
Oversights in Clinical Translation: The paper does a great job of reviewing the scientific aspects of liver tissue engineering, but it does not sufficiently address clinical implications. There needs to be some mention of how these tissue-engineered livers would function in vivo, the potential complications, and the current limitations in translating these technologies to the bedside. These considerations are key towards the clinical relevancy of the research
-
The manuscript has many complicated sentences that could be more reader-friendly if they were broken down into simpler sentences. Consider revising to improve the readability and coherence of the material.
-
Language and Grammar: I noticed a few minor typos like 'sectrion', 'defenetration' etc. You may want to fix these as well as properly spell-check and proof-read the article to avoid such small errors.
-
Overall, your manuscript is strong and informative, offering solid research on the topic. After correcting minor errors and final proofreading, it should be ready for submission.
I hope you find these comments helpful! Please do not hesitate to reach out with any other questions or concerns.
Comments on the Quality of English LanguageLong sentences doesn't make meaning. Spelling errors at many places.
Author Response
Dear Reviewer 1,
please see our responses to your valuable comments in the attached pdf-file.
Regards
Ralf Weiskirchen

Reviewer 2 Report
Comments and Suggestions for Authors
In this review the authors have collected some of the recent knowledge related to liver tissue engineering as an alternative method to liver transplantation. Although the review is quite well written, it doesn't add much information to existing data. Similar reviews can be found in literature on the topic.
I believe that one of the important aspects in evaluating reviews is originality. As reported in the revision, the paper does not reach sufficient levels of innovation. There are many reviews in the literature on the aspects covered by this one.
Also the graphics is poor in comparison with that which can be found in literature and could be implemented and improved. For these reasons I don't think this review is acceptable for publication. One possibility is that the authors could go into details about some aspects (nanotechnologies for example), and send a new version of the review totally rewritten and reorganized.
Minor editing of English language required
Author Response
Dear Reviewer 2,
please see our responses to your valuable comments in the attached pdf-file.
Regards
Ralf Weiskirchen

Reviewer 3 Report
Comments and Suggestions for Authors
Liver tissue engineering aims to generate functional liver tissue in vitro that can supplement liver transplantation for treating end-stage liver diseases. Major technical hurdles include obtaining viable liver cell sources, maintaining hepatic phenotype in culture conditions, and vascularizing tissue constructs. Approaches utilize biomaterial or decellularized scaffolds seeded with primary hepatocytes or stem cell-derived liver cells to recreate tissue architecture. Optimization would enhance understanding of liver regeneration pathways and growth factors like HGF and TNF-alpha. Advances in organ-on-a-chip microfluidic culture systems, 3D bioprinting, anti-inflammatory biomaterials, and integration of artificial intelligence systems hold future promise for realizing clinical potential of lab-grown, transplantable liver tissue.
However, a few concerns need to be concisely addressed: the authors should enhance the maturity of stem cell-derived hepatocytes using refined differentiation protocols optimized for hepatic phenotype and function; detailed molecular profiling of cell-scaffold interactions would enable tailoring scaffold properties to stabilize seeded liver cell behavior; incorporating tunable delivery systems for regenerative signals like HGF can better replicate the dynamic liver milieu; finally, incorporating sensors and modeling approaches would allow evaluation of physiological and pathological relevance relative to human liver. Addressing these key aspects around cell sourcing, molecular signaling, and quantitative assessment would propel this work to become more therapeutically translatable and clinically predictive for supplementing liver transplantation capabilities.
Comments on the Quality of English Languagenone.
Author Response
Dear Reviewer 3,
please see our responses to your valuable comments in the attached pdf-file.
Regards
Ralf Weiskirchen

Round 2
Reviewer 1 Report
Comments and Suggestions for Authors
Thank you for addressing the comments.
Reviewer 2 Report
Comments and Suggestions for Authors
I have no issues